# Galactosylceramide Upregulates the Expression of the *BCL2* Gene and Downregulates the Expression of *TNFRSF1B* and *TNFRSF9* Genes, Acting as an Anti-Apoptotic Molecule in Breast Cancer Cells

**DOI:** 10.3390/cancers16020389

**Published:** 2024-01-17

**Authors:** Jaroslaw Suchanski, Safoura Reza, Anna Urbaniak, Weronika Woldanska, Bartlomiej Kocbach, Maciej Ugorski

**Affiliations:** Department of Biochemistry and Molecular Biology, Wroclaw University of Environmental and Life Sciences, C. K. Norwida 31, 50-375 Wroclaw, Poland; jaroslaw.suchanski@upwr.edu.pl (J.S.);

**Keywords:** breast cancer, UGT8, glycosphingolipids, galactosylceramide, apoptotic gene regulation, chemoresistance

## Abstract

**Simple Summary:**

Galactosylceramide (GalCer) increases the resistance of breast cancer to anti-cancer drugs by acting as an anti-apoptotic molecule. This glycosphingolipid was found to specifically downregulate the levels of the pro-apoptotic *TNFRSF1B* and *TNFRSF9* genes and upregulate the levels of the anti-apoptotic *BCL2* gene, suggesting that GalCer regulates their expression at the transcriptional level. Consistent with this hypothesis, (1) MDA-MB-231 and MCF7 cell lines with increased GalCer content showed low activity of the *TNFRSF1B* and *TNFRSF9* promoters; (2) in such MCF7 cells, the activity of the BCL2 promoter was high. However, no differences in *BCL2* promoter activity were observed between MDA-MB-231 cells with low and high GalCer contents. Instead, GalCer increased the stability of Bcl-2 mRNA in the latter cells. The regulatory protein that simultaneously increases the expression of the *TNFRSF1B* and *TNFRSF9* genes and decreases the expression of the *BCL2* gene and the stability of Bcl-2 transcripts is most likely P53, the expression of which is GalCer dependent.

**Abstract:**

Galactosylceramide (GalCer) increases the resistance of breast cancer cells to doxorubicin, paclitaxel, and cisplatin by acting as an anti-apoptotic molecule. GalCer was found to specifically downregulate the levels of the pro-apoptotic *TNFRSF1B* and *TNFRSF9* genes and upregulate the levels of the anti-apoptotic *BCL2* gene, suggesting that this glycosphingolipid regulates their expression at the transcriptional level. Consistent with this hypothesis, MDA-MB-231 and MCF7 breast cancer cells with high levels of GalCer showed lower activity of the *TNFRSF1B* and *TNFRSF9* promoters than cells lacking GalCer. In contrast, the activity of the *BCL2* promoter was higher in MCF7 cells overproducing GalCer than in MCF7 cells without GalCer. However, no difference in *BCL2* promoter activity was observed between MDA-MB-231 cells with high and no GalCer content. Instead, we found that high levels of GalCer increased the stability of Bcl-2 mRNA. Subsequent studies showed that breast cancer cells with high levels of GalCer are characterized by significantly lower expression of P53. Importantly, inhibition of P53 expression by siRNA in MCF7 and MDA-MB-231 cells lacking GalCer resulted in decreased expression and promoter activity of the *TNFRS1B* and *TNFRSF9* genes. On the other hand, increased expression and promoter activity of the *BCL2* gene was found in such MCF7 cells, and increased stability of Bcl-2 transcripts was observed in such MDA-MB-231 cells. Taken together, these data strongly suggest that the regulatory protein that simultaneously increases the expression of the *TNFRSF1B* and *TNFRSF9* genes and decreases the expression of the *BCL2* gene and the stability of Bcl-2 transcripts is most likely P53, the expression of which is GalCer dependent.

## 1. Introduction

Breast cancer (BC) is one of the greatest oncological challenges in developed countries. High morbidity, together with a high mortality rate, makes BC exceptionally important medically. Drug resistance (DR) is one of the major concerns, particularly in patients with advanced cancer. Different mechanisms of DR are responsible for tumor treatment failures, including decreased intracellular drug concentrations, disturbances in the cell cycle, resistance to apoptosis, DNA repair deficiencies, activation of intracellular pathways related to cancer progression, epigenetic alterations, and changes in drug targets [1]. One class of cellular compounds involved in the DR of cancer cells is glycosphingolipids (GSLs) [2]. The involvement of GSLs in DR was first described by Lavie et al. [3], who showed that DR cancer cells were characterized by the accumulation of glucosylceramide (GlcCer). Initially, it was proposed that increased synthesis of GlcCer highly decreases the intracellular content of pro-apoptotic ceramides, making cancer cells more resistant to drug-induced apoptosis [4]. However, further studies showed that overexpression of GCS and accumulation of GlcCer correlated with high expression of P-glycoprotein (Pgp, MDR1), suggesting that GlcCer positively regulates the expression of the *MDR1* gene [5]. A subsequent study revealed that increased levels of globo-series GSLs (globotriaosylceramide and globotetraosylceramide), but not GlcCer by itself, are responsible for up-regulation of Pgp in cancer cells [6,7,8]. In addition to globo-series GSLs, a high content of GM3 ganglioside was shown to protect murine lung carcinoma cells from cytotoxic effects of etoposide and doxorubicin [9] and small-cell lung cancer cells from cytotoxic effects of DOX and cisplatin [10]. However, in chronic myeloid leukemic K562 cells, increased GM3 content decreased resistance to etoposide and staurosporine [11]. In the case of GM3, increased or decreased resistance to chemotherapy has been linked to differences in the resistance of cancer cells to apoptosis induced by anti-cancer drugs.

Galactosylceramide (GalCer) is a GSL synthesized by galactosylceramide synthase (UGT8, C.E. 2.4.2.62) [12] encoded by the *UGT8* gene [13] and is a major myelin-stabilizing component [14]. However, it was recently shown that BC cells expressing high levels of UGT8 and accumulating GalCer form tumors and metastatic colonies in the lungs more efficiently than BC cells with suppressed UGT8 expression [15]. In accordance with these results, tumor specimens with elevated GalCer content were characterized by higher Ki-67 proliferation index values and a reduction in the number of apoptotic cells. Therefore, it was hypothesized that GalCer serves as an anti-apoptotic molecule, and its presence increases the ability of cancer cells to survive in the unfavourable tumor microenvironment [15,16]. In addition, GalCer was found to increase resistance of BC cells to doxorubicin, suggesting that GalCer can be an important factor in the BC cells’ resistance to chemotherapy. This hypothesis is endorsed by findings that GalCer content is increased in DR colon and ovarian cancer cells [17,18] and that Krabbe cells synthesizing large amounts of GalCer are more resistant to apoptosis induced by daunorubicin and cytosine arabinoside than Gaucher cells, which contain low levels of this GSL [19]. Furthermore, inhibition of GlcCer in U937 and HL60 cells with such inhibitors as D,L-threo-1-phenyl-2-palmitoylamino-3-morpholino-1-propanol (PDMP) or 1-phenyl-2-palmitoylamino-3-morpholino-1-propanol (PPMP), which was accompanied by increased levels of GalCer, protects them from daunorubicin-induced apoptosis [19]. In addition, such cells with high contents of exogenously added GalCer became more resistant to daunorubicin-induced apoptosis. Accumulation of GalCer was observed in Madin–Darby canine kidney cells exposed to hyperosmotic and heat stress [20,21]. Based on this data and the generally accepted idea that ceramide is one of the important pro-apoptotic molecules [22,23], we proposed that the anti-apoptotic activity of GalCer could be linked to decreased levels of ceramide due to increased synthesis of GalCer [15]. However, it was found that BC cells with various amounts of GalCer contain essentially the same amounts of ceramide [16], suggesting that glycosylation of ceramide is not a major mechanism that increases resistance of tumor cells to drug-induced apoptosis. Consequently, a combination of molecular and pharmacological techniques were used in this study to examine the function of GalCer in BC cell resistance to anti-cancer agents and elucidate the mechanisms of its anti-apoptotic activity.

## 2. Materials and Methods

### 2.1. Cell Lines and Culture Conditions

Human BC MDA-MB-231 and T47D cells were obtained from the American Type Culture Collection (ATCC, Manassas, VA, USA). The BC MCF7 and murine 4T1 mammary cancer cell lines were kindly provided by the Cell Line Collection of the Hirszfeld Institute of Immunology and Experimental Therapy (Wroclaw, Poland). MCF7 cell line was authenticated by the ATCC Cell Line Authentication Service. T47D.UGT8 cells overexpressing UGT8 and control T47D.C cells derived from T47D cells have previously been described by Suchanski et al. [16]. MDA-MB-231 and MCF7 cells were cultured in α-MEM supplemented with 10% fetal bovine serum (FBS, Cytogen, Krakow, Poland), 2 mM L-glutamine, and antibiotics. T47D and 4T1 cells were cultured in RPMI-1640 Medium with 10% FBS (Cytogen, Krakow, Poland), 2 mM L-glutamine, and antibiotics. For T47D cells, the culture medium was supplemented with bovine insulin (0.2 U/mL) (Merck, Darmstad, Germany).

### 2.2. Vector Construction, Virus Production, and Transductions

Using PCR, human UGT8 cDNA was generated from the MDA-MB-231 cDNA library as a template for the primers forXhoI-UGT8 and revBamHI-UGT8 (Appendix A), and mouse galactosylceramide synthase (UGT8a) cDNA was amplified from the pCMV6-Entry vector containing UGT8a cDNA (MR208632, 1626 bp; OriGene, Rockville, MD, USA) with forEcoRI-UGT8a and revMluI-UGT8a primers (Appendix A). To generate UGT8 and UGT8a-expressing vectors, the resulting inserts were cloned into a pLVX-Puro Vector (Clontech, Mountain View, CA, USA) as previously described [16] and named pLVX-UGT8-Puro and pLVX-UGT8a-Puro, respectively. For knockout of the *UGT8* gene, a *UGT8* CRISPR/Cas9 vector encoding the Cas9 nuclease and UGT8-specific 20 nucleotide guide RNA (gRNA-TGTGATAGCTCATCTTTTAG) was purchased from Applied Biological Materials Inc. (Vancouver, BC, Canada). For the generation of lentivirus, LentiX 293T cells (Clontech, Mountain View, CA, USA) were transfected at 50–60% confluence with Lenti-X Packaging Single Shots (Takara, Shiga, Japan). The production of virus particles and cell transductions have been previously described [15]. *UGT8* knockout cells were selected using puromycin (1 μg/mL) (Thermo Fisher Scientific, Oxford, UK). Antibiotic-resistant cells were subcloned using the limiting dilution technique. The mutation introduced by CRISPR-Cas9 editing was identified by sequencing genomic DNA with the primers F1-exon2/UGT8 and R1-exon2/UGT8 (Appendix A).

### 2.3. Quantitative PCR (qPCR) Assay and Apoptosis Gene Expression Profiling with RT-qPCR Array

RNA was purified from BC cells using an RNeasy Mini Kit (Qiagen, Hilden, Germany). To synthesize cDNA, the SuperScipt RT Kit (Thermo Fisher Scientific, Oxford, UK) was used. The relative amounts of mRNAs were quantified through qPCR with iQ SYBR Green Supermix (Bio-Rad, Hercules, CA, USA) using an iQ5 Optical System (Bio-Rad, Hercules, CA, USA) and *ACTB* as the reference gene. The primers for qPCR are shown in Appendix A. The gene expression profiles of 46 apoptosis-related genes were determined through RT-qPCR (Appendix A) using the EvaGreen dye-based detection system (Biotium, Fremont, CA, USA). Two reference genes, *ACTB* and *GAPDH*, were used to normalize RNA levels. PCR amplification consisted of an initial denaturation for 3 min at 95 °C, followed by 35 cycles of 20 s denaturation at 95 °C, annealing for 20 s at 5 °C, and extension for 20 s at 72 °C. The relative changes in gene expression were calculated using the ΔΔCt (cycle threshold) method. Fold change values were calculated using the 2^−ΔΔCt^ formula.

### 2.4. Western Blotting

Western blotting was performed as previously described [16]. The following rabbit polyclonal antibodies were used to detect specific proteins: anti-UGT8 (Biorbyt, UK), anti-Bcl-2 (Abcam, Cambridge, UK), anti-TNFRSF1B (Merck, Darmstad, Germany), anti-TNFRSF9 (Biorbyt), anti-P53 (Cell Signaling, Danvers, MA, USA), anti-CREB (Cell Signaling, Danvers, MA, USA), anti NF-κB p65 (Abcam, Cambridge, UK), and mouse monoclonal anti-GAPDH antibody (Novus Biologicals, Cambridge, UK).

### 2.5. Purification of Neutral GSLs

GSLs were extracted from 10^8^–10^9^ cells using the chloroform–methanol extraction method and neutral GSLs were purified as previously described [24].

### 2.6. Thin-Layer Chromatogram (TLC)-Binding Assay

High-performance-thin layer chromatography (HP-TLC) was performed on silica gel 60 HP-TLC plates (Merck, Darmstad, Germany). Neutral GSLs were separated using a 2-isopropanol:methyl acetate:15 M ammonium hydroxide:water solvent system (75:10:5:15, *v*/*v*/*v*/*v*). GSLs were visualized by spraying the plate with primuline reagent (0.05% primuline in acetone/water, 4:1 *v*/*v*) and heating for 1 min at 120 °C. GalCer was detected by a TLC-binding assay as described previously [15], using rabbit polyclonal anti-GalCer antibodies (Merck, Darmstad, Germany).

### 2.7. Cell Survival Assay

The viability of the cells was assessed through an MTT assay. Briefly, 5 × 10^3^ cells/well were seeded in a 96-well plate (Nunc, Rochester, NY, USA) and incubated with increasing concentrations of doxorubicin hydrochloride (DOX, Pfizer, New York, NY, USA), paclitaxel (PAX, Ebewe Pharma, Attersee, Austria), or cisplatin cis-diamminedichloroplatinum (II) (CDDP, Merck) for 48 h. Then, the medium was discarded, and the cells were incubated with MTT solution (0.5 mg/mL, Merck) for 4 h at 37 °C. After removing the MTT solution, 200 μL of DMSO (Chempur, Piekary Śląskie, Poland) was added for 15 min at 22 °C and the optical density was measured at 570 nm.

### 2.8. Apoptotic Assay

The cells (5 × 10^5^) were seeded in multi-well plates (Nunc). The following day, the cells were treated with DOX with or without the selective inhibitor ABT-199 of Bcl-2 protein (Cayman Chemical, Ann Arbor, MI, USA) or TNFα (Thermo Fisher Scientific) for 48 h. The percentages of apoptotic cells were determined using the Dead Cell Apoptosis Kit (Thermo Fisher Scientific). Fluorescence was measured on a BD FACS Lyric (Becton-Dickinson, Franklin Lakes, NJ, USA) and data were processed using the BD FACSuite™ Software v1.5.0.925 (Becton-Dickinson). The cells in early apoptosis were stained with Annexin V APC, whereas cells in late apoptosis were stained with Annexin V APC and SYTOX Green.

### 2.9. In-Vivo Tumor Growth Assay

The animal experiments were approved by the Second Local Ethics Committee for Animal Experimentation (033/2019/P1, Wroclaw, Poland). Six-week-old female BALB/c mice were obtained from the Mossakowski Medical Research Institute, Polish Academy of Sciences (Warsaw, Poland) and kept under specific pathogen-free conditions at room temperature. Suspensions of 4T1 cells (2 × 10^4^ cells/100 µL PBS) were mixed with an equal volume of ice-cold BD Matrigel Matrix High Concentration (Becton-Dickinson) and injected subcutaneously (s.c.). Growth of tumors was monitored once a week by measuring the tumor diameter using a caliper. The tumor volume (TV) was calculated as TV (mm^3^) = (d2 × D)/2, where d is the shortest diameter and D is the longest diameter. For the therapeutic experiments, mice were injected intravenously (i.v.) twice a week with DOX (1.5 mg/kg body weight). Mice were euthanized via cervical dislocation following light anaesthesia through isoflurane (Forane, Abott Laboratories, North Chicago, IL, USA) inhalation. Tissue samples were collected in 10% buffered formalin and subjected to histological analyses.

### 2.10. Cloning of Human TNFRSF1B, TNFRSF9, and BCL2 Gene Promoters and Determination of Their Activity Using Luciferase Reporter Vectors

The Genomatix MatInspector software version 2.7 (http://www.genomatix.de (accessed on 1 October 2021)) was applied to identify the nucleotide sequences of *TNFRSF1B, TNFRSF9,* and *BCL2* promoters, which were generated via PCR using genomic DNA purified from MDA-MB-231 cells as a template and primers listed in Appendix A. They were amplified as follows: 3 min of initial denaturation at 95 °C, followed by 30 cycles of 20 s denaturation at 95 °C, annealing at 55 °C for 20 s, and 1 min of final extension at 72 °C. The resulting PCR products containing additional sequences corresponding to MluI and HindIII restriction sites were cloned into pGL3-Basic luciferase vector (Promega, Madison, WI, USA). The obtained constructs were named pGL3-*TNFRSF1B*/LUC, pGL3-*TNFRSF9*/LUC, and pGL3-*BCL2*//LUC for the *TNFRSF1B, TNFRSF9*, and *BCL2* promoters, respectively.

To determine the activity of the promoters, pGL3-TNFRSF1B/LUC, pGL3-TNFRSF9/LUC, and pGL3-BCL2/LUC constructs (2 μg) were co-transfected with control *Renilla* luciferase-expressing pRL-TK vector (2 μg) using polyethylenimine (Merck) into BC cells seeded into multi-well plates and grown to 70% confluence. After 48 h, to measure promoter activity, cells were analyzed using the Dual-Luciferase Reporter Assay System (Promega) following recommendations provided by the manufacturer.

### 2.11. mRNA Stability Assay

Cells (5 × 10^5^) were treated with 2 μg/mL of actinomycin D (Thermo Fisher Scientific) in DMSO for 2, 4, and 6 h at 37 °C. At the indicated time points, RNA was extracted using Fenozol reagent (A&A Biotechnology, Gdansk, Poland) and the level of Bcl-2 transcripts was analyzed using RT-qPCR with iQ SYBR Green Supermix (Bio-Rad, Hercules, CA, USA) as described above in qPCR assay section.

### 2.12. siRNA Transfections

Transfection with a mixture of P53 siRNAs (Santa Cruz Biotechnology, Dallas, TX, USA) was performed using Lipofectamine RNAiMAX transfection reagent (Thermo Fisher Scientific). Briefly, cells (2.5 × 10^5^) were seeded in six-well plates in complete α-MEM and, at 60% confluence, incubated with siRNAs for 72 h. The transfection reagent was prepared according to the RNAiMAX Transfection Procedure (Thermo Fisher Scientific). Cells were harvested via trypsinization and subjected to further experiments.

### 2.13. Statistical Analysis

All statistical analyses were performed using Prism 5.0 (GraphPad, San Diego, CA, USA). The two-way ANOVA Dunnett’s post-test was used for statistical analysis of in vivo tumor growth experiments. The Mann–Whitney test was used to compare the groups of data that did not meet the assumptions of the parametric test. In all analyses, the results were considered statistically significant at *p* < 0.05.

## 3. Results

### 3.1. DR of BC Cells Depends on Expression of UGT8 and Synthesis of GalCer

We previously showed that GalCer is responsible for the resistance of BC cells to DOX [15]. Based on these results, this study was carried out to further assess the GalCer function in BC cell resistance to commonly used chemotherapeutics, using cells with different expression levels of this GSL. A model, representing a loss-of-function phenotype, was obtained through CRISPR-edited knockout of the *UGT8* gene in MDA-MB-231 cells (Appendix A). The two UGT8-negative, puromycin-resistant clones, named MDA.Δ.UGT8.1 and MDA.Δ.UGT8.4, were chosen for further studies (Figure 1A). In addition, a control cell line, MDA.Δ.C, was created by transducing MDA-MB-231 cells with the CRISPR/Cas9 plasmid encoding a non-specific 20 nt gRNA. The model, representing the gain-of-function phenotype, was obtained overexpressing UGT8 in MCF7 cells (Figure 1B). The cells that accumulated GalCer were named MCF7.UGT8. The control MCF7.C cell line was generated by transducing MCF7 cells with an empty vector. A second gain-of-function cellular model used in this study was represented by T47D.UGT8 cells [16]. All cells used in these studies had the same morphology and proliferative potential as parental cells (Appendix A).

To assess, whether the presence of GalCer affects the viability of BC cells treated with chemotherapeutic agents, cells were incubated with DOX, PAX, and CDDP at different concentrations. Using loss-of-function and gain-of-function cellular models, it was shown that all analyzed GalCer-rich BC cells were more resistant to cytotoxic effects induced by DOX, PTX, and CDDP than BC cells unable to synthesize this GSL (Figure 1C). Notably, the sensitivity of the control MDA.Δ.C, MCF7.C, and T47D.C cells to anti-cancer drugs was the same as the respective parental wild-type BC cells.

### 3.2. Resistance of Murine 4T1 Mammary Tumours to DOX Treatment Is Dependent on the Presence of GalCer

To further prove that GalCer is directly involved in increased resistance of BC cells to anticancer drugs, an in vivo study using murine 4T1.UGT8a mammary carcinoma cells overexpressing mouse UGT8a and GalCer was performed (Figure 2A,B). The 4T1.UGT8a cells were obtained by transducing 4T1 cells with a vector encoding mouse UGT8a cDNA. Similar to human BC cells, 4T1.UGT8a cells were more resistant to DOX as their viability was higher than that of control 4T1.C cells carrying the “empty” vector (Figure 2C). Initially, we assessed whether the presence of GalCer affected the tumorigenicity of murine mammary carcinoma cells. After s.c. transplantation into BALB/c mice, on the day 22, the mean tumor volumes (357 ± 63 mm^3^) formed by 4T1.UGT8a cells were similar to tumor volumes (373 ± 68 mm^3^) formed by control 4T1.C cells (Figure 2D). To further evaluate, whether the sensitivity of these tumors to DOX treatment can be affected by the accumulation of GalCer, mice were injected intravenously on day 7, 10, 13, and 16 after mammary carcinoma cell transplantation with DOX (Figure 2D).

Tumor regression was observed in mice transplanted with 4T1.UGT8a cells with high GalCer and GalCer-negative 4T1.C cells compared to mice treated with placebo. However, only in the case of GalCer-negative 4T1.C tumors was the reduction in tumor volume statistically significant in comparison to that in mice subjected to treatment with placebo. Therefore, these data confirm our proposal that GalCer is involved in DR of BC tumors.

### 3.3. GalCer Is Responsible for Changes in the Expression of Specific Apoptotic Genes

To further elucidate the mechanism by which GalCer mediates cytoprotective effects on BC cells, we compared the expression of 46 apoptotic genes in BC cells with high and low GalCer content using qPCR (Appendix A).

All analyzed BC cells that had a high content of GalCer (MDA.Δ.C, MCF7.UGT8, and T47D.UGT8) were characterized by considerably increased levels of the anti-apoptotic *BCL2* gene and substantially decreased levels of pro-apoptotic *TNFRSF1B* and *TNFRSF9* genes in comparison to BC cells with low content of GalCer (MDA.Δ.UGT8.1, MDA.Δ.UGT8.4, MCF7.C, and T47D.C). Consistent differences between BC cells with high and low content were found for these three genes, and their expression in loss-of-function and gain-of-function cellular models was further validated at the mRNA and protein levels (Figure 3A,B). As expected, the expression of Bcl-2 protein was markedly increased in MDA.ΔC, MCF7.UGT8, and T47D.UGT8 cells compared to MDA.Δ.UGT8.1, MDA.Δ.UGT8.4, MCF7.C, and T47D.C cells. In contrast, the levels of TNFRSF1B and TNFRSF9 proteins were higher in MDA.Δ.UGT8.1, MDA.Δ.UGT8.4, MCF7.C, and T47D.C cells than in MDA.Δ.C, MCF7.UGT8, and T47D.UGT8 cells.

To prove that the changes in Bcl-2 expression observed in BC cells with various contents of GalCer directly affects their sensitivity to anti-cancer drugs, MDA.C cells with high levels of GalCer and high Bcl-2 expression were treated with ABT-199, which is a selective Bcl-2 inhibitor [25], and subsequently incubated with DOX. Using flow cytometry, it was found that treatment of MDA.C cells with ABT-199 increased their sensitivity to DOX-induced apoptosis compared to MDA.C cells incubated with DOX only (Figure 4A, Appendix A). However, the sensitivity of the MDA.Δ.UGT8.4 cells with a lack of GalCer and low levels of Bcl-2 to DOX-induced apoptosis was not affected by treatment with the Bcl-2 inhibitor. These results were verified using a gain-of-function cellular model. When MCF7.UGT8 cells, also with high GalCer content and high Bcl-2 expression, were treated with ABT-199, their sensitivity to DOX-induced apoptosis was markedly increased compared to MCF7.UGT8 cells incubated with DOX only (Figure 4B, Appendix A). In contrast, DOX-induced apoptosis was not affected by the Bcl-2 inhibitor in GalCer-negative MCF7.C cells with low Bcl-2 levels.

The effects of TNFRSF1B and TNFRSF9 on BC cell apoptosis were assessed by treating cells with human recombinant TNFα and flow cytometry. MDA.Δ.UGT8.4 cells with *UGT8* gene knockout increased the expression of TNFRSF1B and TNFRSF9 and were more prone to apoptosis triggered by TNFα than control MDA.Δ.C cells (Figure 4C, Appendix A). However, in MCF7.UGT8 cells overexpressing GalCer with low levels of TNFRSF1B and TNFRSF9 expression that were incubated with TNFα, the percentage of apoptotic cells was considerably lower than in control MCF7.C cells (Figure 4D, Appendix A).

In summary, we found that GalCer-driven changes in the expression of Bcl-2, TNFRSF1B, and TNFRSF9 were responsible for the sensitivity/resistance of BC cells to DOX-induced apoptosis.

### 3.4. GalCer Regulates the Expression of TNFRSF1B, TNFRSF9, and Bcl-2 on the Level of Transcription and mRNA Stability

Changes in the expression of TNFRSF1B, TNFRSF9, and Bcl-2 mRNAs found in BC cells with different GalCer levels suggested that this glycosphingolipid may influence the transcription of these genes. Therefore, to address this hypothesis, the *TNFRSF1B*, *TNFRSF9*, and *BCL2* promoters were cloned and sequenced (Appendix A), and a dual-luciferase reporter assay system was used to analyze their activities in BC cells.

The activities of the *TNFRSF1B* and *TNFRSF9* promoters were found to be higher in MDA.Δ.UGT8.4 and MCF7.C cells lacking GalCer than in MDA.Δ.C and MCF7.UGT8 with high GalCer levels (Figure 5A,B). In contrast, in the case of *BCL2*, higher promoter activity was observed in cells with a high GalCer level (Figure 5C). However, no differences in *BCL2* promoter activity between MDA.Δ.C and MDA.Δ.UGT.4 were observed. Based on this result, we hypothesized that in MDA-MB-231 cells, Bcl-2 expression is regulated at the post-transcriptional level as shown in HL60, Jurkat, or HEK 293 cells [26]. One of these mechanisms entails the control of mRNA stability [27]. We therefore analysed Bcl-2 transcript levels in MDA.Δ.C and MDA.Δ.UGT.4 cells treated with actinomycin D and found that the stability of Bcl-2 mRNA was significantly higher in control MDA.Δ.C cells than in MDA.Δ.UGT.4 cells with blocked GalCer synthesis (Figure 5D).

Based on these results, an in silico analysis was performed to identify the binding sites for potential transcription factors affecting the activities of *TNFRSF1B*, *TNFRSF9*, and *BCL2* promoters in BC cells. Using the Genomatix MatInspector software, numerous transcriptional elements were identified in each of the analyzed promoter sequences. However, only the binding sites for the transcription factors: P53, NF-ĸB1, and CREB1 were present in all three promoters (Appendix A), suggesting that they may regulate *TNFRSF1B*, *TNFRSF9*, and *BCL2* gene expression. Therefore, we analyzed the expression of CREB1, NF-ĸB1, and P53 in BC cells with different levels of GalCer using Western blotting. MDA.Δ.C cells with a high level of GalCer and high expression level of Bcl-2 expressed significantly less P53 than MDA.Δ.UGT.4 cells lacking GalCer and with low expression levels of Bcl-2 (Figure 6A). In contrast, GalCer-negative MCF7.C cells with low levels of Bcl-2 are characterized by considerably higher expression of P53 than MCF7.UGT8 cells overexpressing GalCer. On the other hand, no differences in the expression of CREB1 and NF-ĸB1 were found between MDA-MB-231 and MCF7 cells with different expression of GalCer (Appendix A). Furthermore, CREB1 and NF-ĸB1 upregulate the expression of TNFR family members as well as a Bcl-2 [28,29,30,31,32]. In contrast, P53 upregulates *TNFRSF1B* and *TNFRSF9* genes but downregulates the expression of the *BCL2* gene [33]. Therefore, only P53 meets the conditions for a regulatory factor affecting simultaneous expression of *TNFRSF1B*, *TNFRSF9*, and *BCL2* genes, but in the opposite way. To prove that changes in the expression of *TNFRSF1B* and *TNFRSF9* genes are directly related to changes in the expression of P53, MDA.Δ.UGT.4 and MCF7.C cells were subjected to treatment with P53 siRNAs.

Inhibition of P53 expression in MDA.Δ.UGT.4 as well as MCF7.C cells resulted in highly decreased expression of *TNFRSF1B* and *TNFRSF9* genes (Figure 6B). On the other hand, treatment of MCF7.C cells with P53 siRNAs resulted in increased expression of the *BCL2* gene, and a lack of differences in expression of *BCL2* gene was found in P53 siRNA-treated MDA.Δ.UGT.4 cells. To further confirm that P53 directly regulates the expression of *TNFRSF1B*, *TNFRSF9*, and *BCL2* genes at the transcriptional level, we analyzed promoter activities in BC cells after inhibition of P53 expression using siRNA. In agreement with previous findings (see Figure 5), decreased P53 levels resulted in highly decreased activity of *TNFRSF1B* and *TNFRSF9* promoters and increased activity of *BCL2* promoter in MCF7.C cells, as well as decreased activity of *TNFRSF1B* and *TNFRSF9* promoters in MDA.Δ.UGT.4 cells (Figure 6C). Additionally, in line with previous observations, inhibition of P53 expression increased the stability of the Bcl-2 transcript (Figure 6D).

## 4. Discussion

We previously showed that BC MDA-MB-231 cells, which synthesize large amounts of GalCer, become more resistant to DOX [15]. In the present study, using three human BC cellular models with loss-of-function (MDA-MB-231 cells) or gain-of-function (MCF7 and T47D cells) phenotypes, we extended this finding and proved that BC cells are also more resistant to other anti-cancer drugs such as PTX and CDDP. These in vitro results are strongly supported by an in vivo study using a mouse model. Interestingly, not only human, but also murine mammary carcinoma 4T1 cells overexpressing murine UGT8a and accumulating GalCer become more resistant to DOX. When such cells were injected into BALB/c mice, they showed increased resistance to DOX therapy than the control 4T1 cells. Taken together, these data strongly suggest that GalCer is a marker of DR in BC cells and a new target for breast cancer treatment. In agreement with the general view that the majority of anticancer drugs induce apoptosis in cancer cells [34], we also showed that GalCer serves as an anti-apoptotic molecule, increasing the resistance of BC cells to apoptosis induced by DOX [15].

Based on the above findings, the present study was undertaken to further define the mechanisms by which GalCer increases the resistance of BC cells to drug-induced apoptosis. We found that GalCer specifically downregulated the expression of *TNFRSF1B* and *TNFRSF9* genes and increased *BCL2* gene expression at the mRNA and protein levels, proving that changes in their expression directly affect the apoptotic properties of BC cells. Therefore, these results showed for the first time that the same species of GSL could produce simultaneous but opposite effects on the extrinsic and intrinsic (mitochondrial) apoptotic pathways. In contrast to GalCer, other GSLs involved in the regulation of apoptotic gene expression belong to the ganglio-series of GSLs and act mainly as pro-apoptotic molecules. One of these gangliosides, GD3, is involved in suppression of *BCL2* gene expression at the transcriptional level via dephosphorylation of AKT and CREB [35]. The forced expression of Bcl-2 protected from GD3-induced apoptosis in T cell lymphoma CEM cells and mouse oligodendrocytes suggested that relevant GD3 targets are under Bcl-2 control [36], which we observed in the case of GalCer, but with an opposite effect. However, GD3, in contrast to GalCer, activates the mitochondrial pathway at the level of the permeability transition pore complex causing cytochrome c release and caspase activation [37]. In addition, GD3 acts as a pro-apoptotic agent by preventing (through a currently unknown mechanism) the nuclear localisation of active NF-κB that suppresses the NF-κB-dependent survival pathway [38]. Sequestration of NF-κB in the cytoplasm protects the activation of genes dependent on NF-κB, sensitizes hepatocytes against apoptotic stimuli such as TNFα, and sensitizes human hepatoma cells to daunorubicin and/or ionizing radiation [39]. In contrast, overexpression of ganglioside sialidase (Neu3) in colon carcinoma cells was associated with increased Bcl-2 and decreased caspase expression, which is another example of apoptosis suppression associated with GSLs metabolism [40]. Increased resistance to drug-induced apoptosis associated with Bcl-2 expression was observed in mouse 3LL Lewis lung cancer cells overexpressing GM3 ganglioside compared with GM3-negative parental cells [9]. In this case, upregulation of Bcl-2 was linked to the post-translational protein modifications.

To further prove that GalCer affects the expression of specific apoptotic genes at the transcriptional level, the promoters of *TNFRSF1B, TNFRSF9,* and *BCL2* genes were cloned, and their activities were determined using MDA-MB-231 and MCF7 cellular models. Consistent with our hypothesis, MDA.Δ.C and MCF7.UGT8 cells, both with high GalCer content, showed lower activity of *TNFRSF1B* and *TNFRSF9* promoters than MDA.Δ.UGT8.4 and MCF7.C cells lacking GalCer. In MCF7.UGT8 cells, activity of the *BCL2* promoter was higher than in MCF7.C cells. In silico analysis performed to identify potential transcription factors that bind to promoter sequences and affect the activities of *TNFRSF1B, TNFRSF9,* and *BCL2* promoters revealed that the binding sites for CREB1, NF-ĸB1, and P53 were present in all three promoters, indicating that they can be simultaneously involved in the regulation of *TNFRSF1B, TNFRSF9,* and *BCL2* gene expression. No data concerning transcription factors involved in the regulation of *TNFRSF1B* and *TNFRSF9* gene are available; however, CREB1, NF-ĸB1, and P53 play significant regulatory roles in the case of other TNFR family members, upregulating their expression [28,29]. In contrast, much more is known about the regulation of *BCL2* by these three transcription factors [30,31,32]. CREB1 and NF-ĸB1 increase [30,41] and P53 downregulates [33] the expression of the *BCL2* gene. With regards to MCF7.UGT8 and T47D.UGT8 cells, the accumulation of GalCer leads to downregulation of *TNFRSF1B* and *TNFRSF9* genes and upregulation of the *BCL2* gene in comparison to MCF7.C and T47D.C cells that do not produce GalCer; therefore, the involvement of P53 in their simultaneous but opposite regulation seems to be probable. This assumption is further supported by data obtained from MDA-MB-231 cells. In this cellular model, no difference in promoter activity of the *BCL2* gene was observed between MDA.Δ.C and MDA.Δ.UGT8.4 cells, despite significant differences in mRNA and protein levels. To explain these apparent contradictions, we assumed, based on the available information, that P53 can be engaged in post-transcriptional regulation of Bcl-2 expression by increasing the Bcl-2 transcript stability [42]. Therefore, an actinomycin D assay was performed on MDA.Δ.UGT8.4 and control MDA.C cells, which revealed that the stability of Bcl-2 mRNA was considerably lower in BC cells that did not express GalCer. These data also revealed for the first time that one species of GSL can regulate the expression of specific genes through different mechanisms.

The involvement of P53 in the regulation of *TNFRSF1B*, *TNFRSF9*, and *BCL2* genes was supported by the Western blot analysis, which revealed that the accumulation of GalCer in BC cells is associated with decreased level of P53. On the other hand, no differences in the expression of CREB1 and NF-ĸB1 were found between MDA-MB-231 and MCF7 cells with different expression of GalCer. These correlational observations are strongly supported by the siRNA-based RNA interference study, in which P53 expression was silenced. Direct evidence that P53 is involved in the regulation of *TNFRSF1B*, *TNFRSF9*, and *BCL2* expression came from the experiments with siRNA-mediated suppression of P53 in MCF7.C and MDA.Δ.UGT8.4 cells. First, it was found that inhibition of P53 strongly reduced the expression of the *TNFRSF1B* and *TNFRSF9* genes. Importantly, the activities of the *TNFRSF1B* and *TNFRSF9* promoters were also greatly reduced in these cells, confirming our proposal that the expression of these genes is regulated at the transcriptional level. Second, inhibition of P53 increased the expression of the *BCL2* gene. In MCF7.C cells, this was due to increased promoter activity. However, P53 inhibition had no effect on *BCL2* gene expression at the mRNA level and its promoter activity in MDA.Δ.UGT8.4 cells. Therefore, in light of our results showing that P53 can regulate Bcl-2 expression by increasing Bcl-2 transcript stability, siRNA-treated MDA.Δ.UGT8.4 cells were subjected to an actinomycin D assay, which showed that P53 is indeed involved in the post-transcriptional regulation of Bcl-2.

Since P53 functions primarily as a transcription factor [43], it is not surprising that in MCF7 cells carrying wild-type P53 [44], regulation of all three target genes takes place at the level of promoter activity. However, in the case of MDA-MB-231 cells carrying a mutant form of P53 [45], only *TNFRSF1B* and *TNFRSF9* genes were regulated at the level of transcription, which was not observed in the case of the *BCl2* gene. These differences can be explained by the studies showing that different P53 variants have an altered promoter activity spectrum [46]. In our case, the R280K mutation did not affect P53 binding to the *TNFRSF1B* and *TNFRSF9* promoters but prevented it from binding to the *BCL2* promoter. Instead, such mutant P53 acquires new properties and binds to the 5’UTR of Bcl-2 mRNA, as was shown in case of the genetic variant of P53 with the substitution of proline by arginine within the proline-rich domain expressed by mucous cells [42].

Studies are in progress to identify the specific mechanisms by which GalCer decreases the level of P53 and in this way subsequently affects *TNFRSF1B*, *TNFRSF9,* and *BCL2* expression. At this stage of our research, we propose the following hypothesis: GalCer localized in lipid rafts interacts with a specific protein, most likely a protein kinase, to trigger a signalling pathway leading to reduced expression of P53, which in turn causes decreased expression of *TNFRSF1B* and *TNFRSF9* and increased expression of *BCL2* genes (Figure 7).

## 5. Conclusions

Our data showed that DR is an intrinsic property of a subpopulation of BC cells overexpressing UGT8 and therefore accumulating GalCer [47], and not a result of chemotherapy, as was observed in the case of GCS and the accumulation of GlcCer in BC cells selected by stepwise exposure to drugs [48]. Furthermore, we showed that GalCer is involved in the regulation of specific apoptotic genes (*TNFRSF1B*, *TNFRSF9*, and *BCL2*) and is the only known GSL that can simultaneously affect extrinsic and intrinsic (mitochondrial) apoptotic pathways. Downregulation of death receptor signalling, which contributes to cancer cell sensitivity to cellular stressors, is in agreement with our proposal that GalCer facilitates tumor cell survival in the hostile tumour microenvironment [15]. However, upregulation of the mitochondrial pathway is mainly responsible for the induction of resistance to cytotoxic drugs, as shown in the present study. Therefore, we propose that the inhibition of UGT8 by knockout of the *UGT8* gene or using chemical inhibitors of the UGT8 enzyme would increase cancer cell sensitivity to conventional chemotherapy.

## Figures and Tables

**Figure 1 cancers-16-00389-f001:**
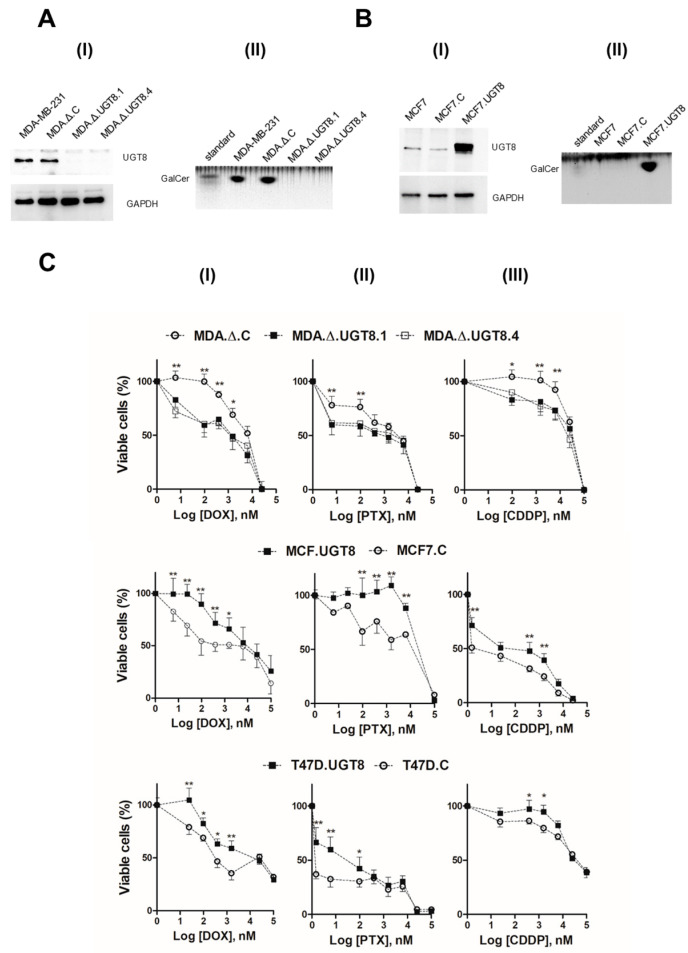
Characteristics of BC cells with knock-out or overexpression of the *UGT8* gene. (**A**) (**I**) Levels of UGT8 protein expression in parental MDA-MB-231 cells, MDA-MB-231 cells transduced with an “empty” vector (MDA.Δ.C) and MDA-MB-231 cell clones with knockout of the *UGT8* gene (MDA.Δ.UGT8.1 and MDA.Δ.UGT8.4). (**A**) (**II**) Immunostaining of neutral GSLs from MDA-MB-231, MDA.Δ.C, MDA.Δ.UGT8.1, and MDA.Δ.UGT8.4 cells. (**B**) (**I**) Level of UGT8 protein expression in parental MCF7 cells, MCF7 cells transduced with an “empty” vector (MCF7.C), and MCF7 cells overexpressing UGT8 (MCF7.UGT8). (**B**) (**II**) Immunostaining of neutral GSLs from MCF7, MCF7.C, and MCF7.UGT8. The expression of UGT8 was analyzed through Western blotting using rabbit polyclonal antibodies directed against UGT8. Cellular proteins (40 µg) separated by SDS-PAGE on 12% gel were transferred onto a nitrocellulose membrane. GAPDH was used as the internal control. For GalCer immunostaining, samples of neutral GSLs that had been purified from 1×10^7^ cells were separated on an HP-TLC plate. (**C**) Viability of BC MDA.Δ.UGT8.1 and MDA.Δ.UGT8.4 cells with knockout of the *UGT8* gene and MCF7.UGT8 and T47D.UGT8 cells overexpressing UGT8 and GalCer exposed to increased concentrations of anti-cancer drugs: doxorubicin (DOX) (**I**), paclitaxel (PTX) (**II**), and cisplatin cis-diamminedichloroplatinum (**III**) (CDDP) (**III**) for 48 h. The percentage of viable cells was determined by MTT assay. Data represent the mean ± standard deviation (SD) of six replicates from two independent measurements. Differences that are statistically significant (* *p* < 0.1, ** *p* < 0.01). The original western blot of Figure 1A,B is in Appendix A.

**Figure 2 cancers-16-00389-f002:**
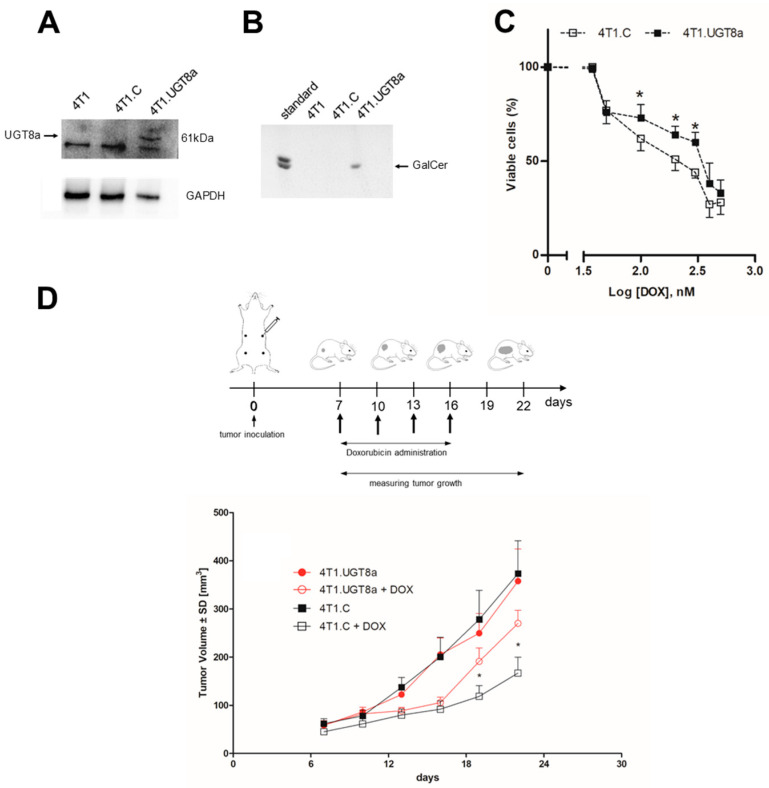
Characteristics of murine mammary carcinoma 4T1 cells overexpressing murine UGT8a and GalCer. (**A**) Expression of UGT8a protein in parental 4T1 cells, control 4T1 cells transduced with vector alone (4T1.C), and 4T1 cells overexpressing UGT8a (4T1.UGT8a); (**B**) immunostaining of neutral GSLs from 4T1, 4T1.C, and 4T1.UGTa cells. Western blotting with anti-human UGT8 antibodies was used to analyze the expression of UGT8a. Cellular proteins (40 µg) separated by SDS-PAGE on 12% gel were transferred onto a nitrocellulose membrane. GAPDH was used as the internal control For GalCer detection, samples of neutral GSLs purified from 1 × 10^7^ cells were applied to an HP-TLC plate. (**C**) Viability of 4T1.C and 4T1.UGT8a cells overexpressing UGT8a and GalCer incubated with increased concentrations of DOX for 48 h. The percentage of viable cells was determined using an MTT assay. Data represent the mean ± SD of six replicates from two independent measurements. Statistically significant differences (* *p* < 0.1). (**D**) Impact of intravenous administration of DOX on the growth of 4T1.C and 4T1.UGT8a tumors in BALB/c mice. 4T1.UGT8a—mice with 4T1.UGT8a tumors overexpressing UGT8a and GalCer treated with placebo (PBS); 4T1.UGT8 + DOX—mice with 4T1.UGT8a tumors treated with DOX; 4T1.C—mice with control 4T1.C tumors treated with placebo (PBS); 4T1.C + DOX—mice with 4T1.C tumours treated with DOX. Data was presented as mean tumor volume (±SD) for the group of mice (*n* = 5) at the specified time intervals. Data was analyzed using the Prism 5.0 two-way ANOVA Dunnett’s post-test (* *p* < 0.05). The original western blot of Figure 2A,B is in Appendix A.

**Figure 3 cancers-16-00389-f003:**
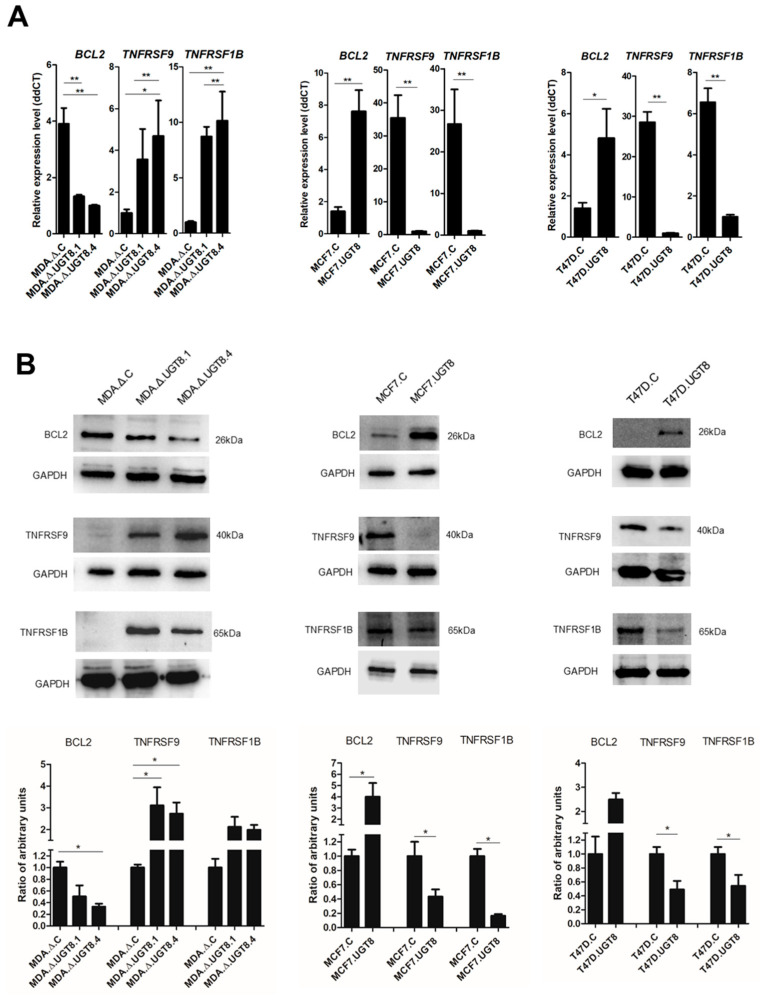
Levels of Bcl-2, TNFRS9, and TNFRS1B expression in BC MDA-MB-231, MCF7, and T47D cells with various contents of GalCer. (**A**) qPCR was used to determine mRNA levels, which were normalized against β-actin mRNA, and corresponding cells with low expression levels of UGT8 mRNA served as calibrator samples. Data is expressed as mean ± SD (* *p* < 0.1, ** *p* < 0.01). (**B**) Western blotting combined with densitometric analysis was used to quantify protein expression levels in BC cell lines. Cellular proteins (40 µg) separated by SDS-PAGE on 12% gel were transferred onto a nitrocellulose membrane. GAPDH was used as the internal control. Band intensities were analyzed with Image Lab software version 6.0.0 build 25 (Bio-Rad Laboratories) and GraphPad Prism version 5. (* *p* < 0.05). All values are means ± SD. The original western blot of Figure 3B is in Appendix A.

**Figure 4 cancers-16-00389-f004:**
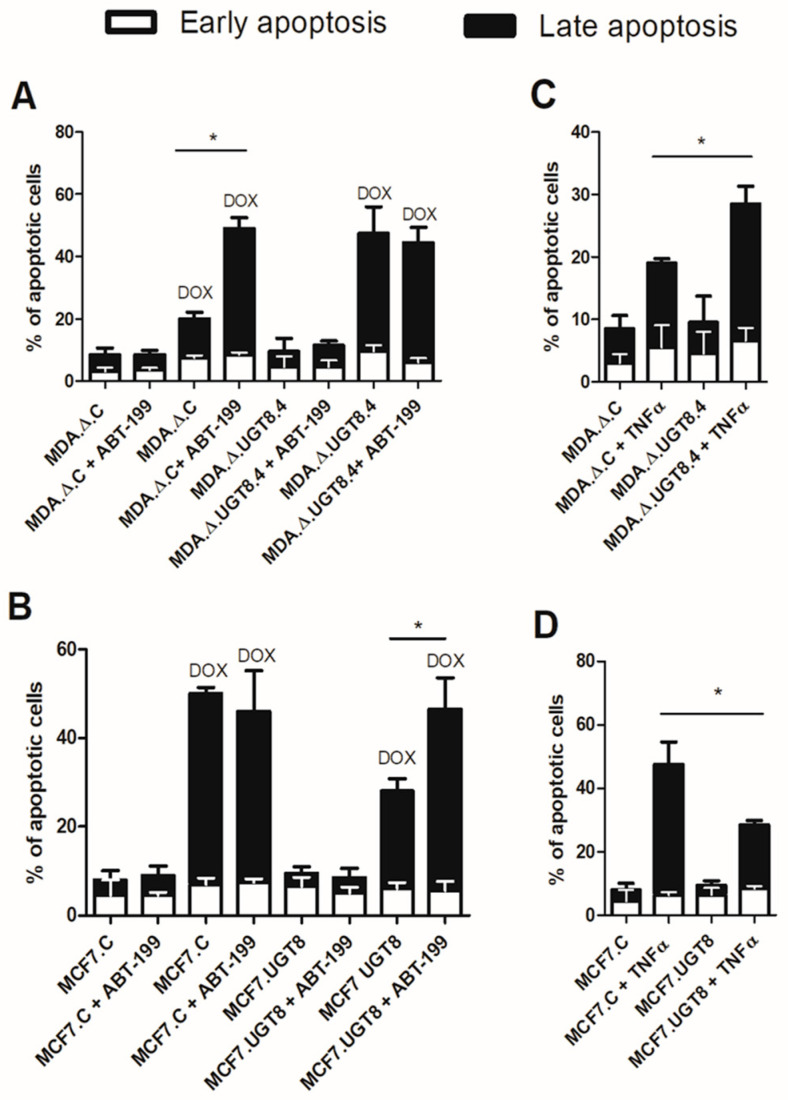
Sensitivity to the DOX-induced apoptosis of BC (**A**) MDA-MB-231 and (**B**) MCF7 cells with various GalCer levels and incubated with ABT-199 (specific inhibitor of Bcl-2). Cells were cultured with ABT-199 at a concentration of 0.125 µM or ABT-199 and DOX at concentrations of 0.125 µM and 1 µM, respectively, for 48 h. Sensitivity to TNFα-induced apoptosis of BC (**C**) MDA-MB-231 and (**D**) MCF7 cells with different GalCer contents. Cells were cultured in the presence of TNFα at a concentration of 25 µM for 48 h. Apoptotic cells were determined through flow cytometry using Annexin V and SYTOX Green stain. MDA.Δ.C—MDA-MB-231 cells transduced with vector alone; MDA.Δ.UGT8.4—MDA-MB-231 cell clones with knockout of the UGT8 gene; MCF7.C—MCF7 cells transduced with vector alone; MCF7.UGT8—MCF7 cells overexpressing UGT8 and GalCer. Data represent the mean ± SD of six replicates from two independent experiments. Statistically significant differences (* *p* < 0.1).

**Figure 5 cancers-16-00389-f005:**
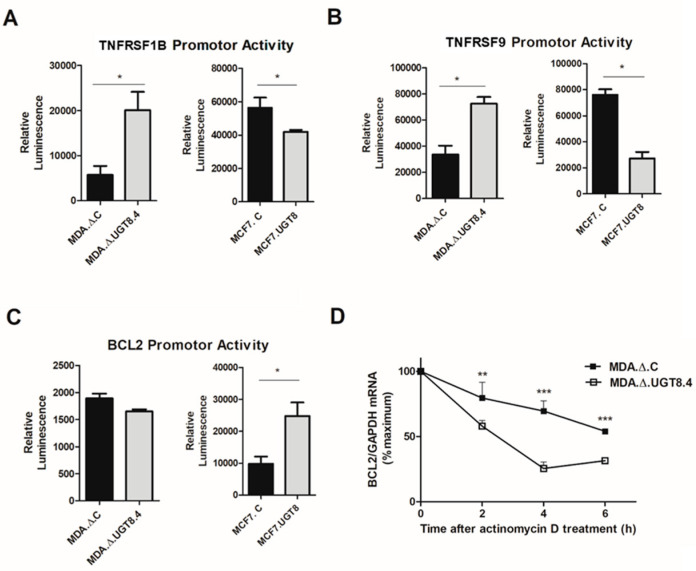
Promoter activities of (**A**) *TNFRS1B*, (**B**) *TNFRS9*, and (**C**) *BCL2* genes cloned into pGL3 basic vector after transfection into BC cells with various contents of GalCer. MDA.Δ.C—MDA-MB-231 cells transduced with an “empty” vector; MDA.Δ.UGT8.4—MDA-MB-231 cell clones with knock-out of the *UGT8* gene; MCF7.C—MCF7 cells transduced with an “empty” vector; MCF7.UGT8—MCF7 cells overexpressing UGT8 and GalCer. The promoter activities were measured using the dual-luciferase reporter assay system. The bars represent average luciferase activities compared with the control pGL3 vector. Data represent the mean ± SD of three replicates from two independent experiments. Statistically significant differences (* *p* < 0.1). (**D**) GalCer increases the stability of Bcl-2 mRNA in BC cells. MDA.Δ.C and MDA.Δ.UGT8.4 cells were treated with actinomycin D for 2—6 h, and after indicated time points, the Bcl-2 mRNA expression levels, which were normalized against β-actin mRNA, were evaluated via qPCR. MDA.Δ.C and MDA.Δ.UGT8.4 cells not treated with actinomycin D were used as calibrator samples. The results are shown as the percentage of the control (0 h). Data represent the mean ± SD of six replicates from two independent experiments. Statistically significant differences (** *p* < 0.01; *** *p* < 0.001).

**Figure 6 cancers-16-00389-f006:**
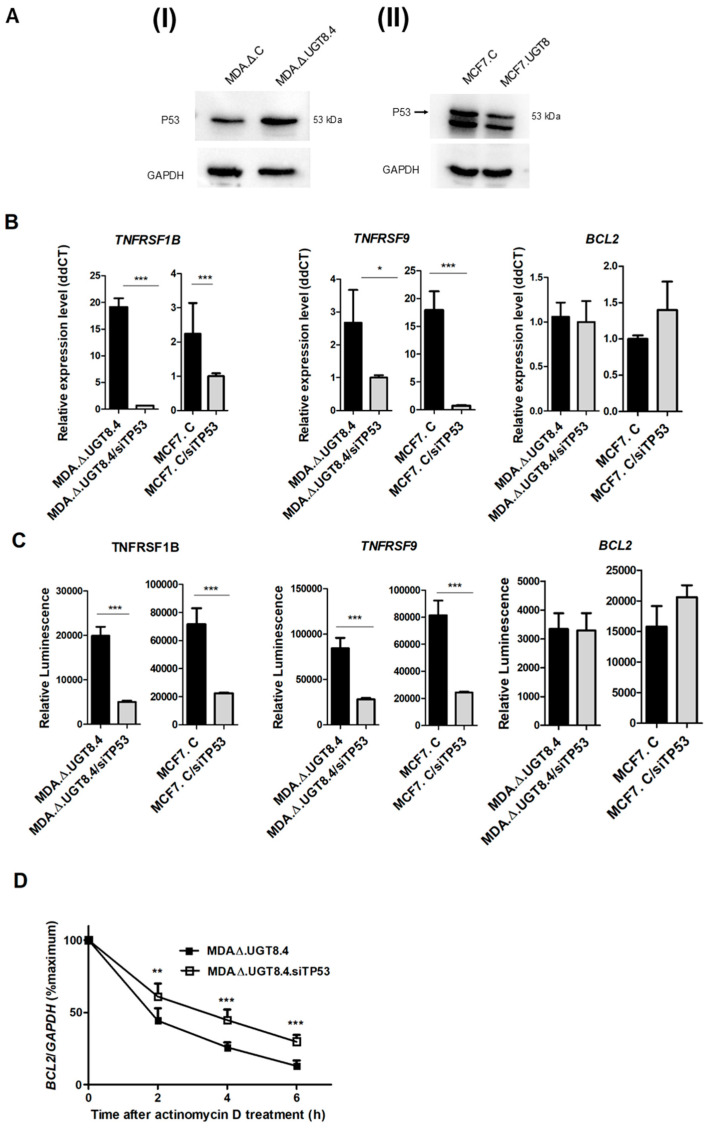
(**A**) P53 expression in MDA-MB-231 cells transduced with an “empty” vector (MDA.Δ.C), MDA-MB-231 cells with knockout of the UGT8 gene and lack of GalCer (MDA.Δ.UGT8.4) (**I**), (**II**) MCF7 cells transduced with a vector alone (MCF7.C), and MCF7 cells overexpressing UGT8 and GalCer (MCF7.UGT8). Western blotting was used to analyze P53 expression in BC cell lines. Rabbit anti-P53 polyclonal antibodies were used to detect P53. Proteins (40 µg) separated by SDS-PAGE on 12% gel were transferred onto a nitrocellulose membrane. GAPDH was used as an internal control. (**B**) *TNFRS1B*, *TNFRS9*, and *BCL2* gene expression levels in MDA.Δ.UGT8.4 and MCF.7 cells transfected with siRNA directed against P53 mRNA. qPCR was used to determine mRNA levels, normalized against β-actin mRNA. Cells with low expression levels of UGT8 mRNA were used as calibrators. Data are expressed as mean ± SD. (**C**) *TNFRS1B*, *TNFRS9*, and *BCL2* promoter activities in MDA.Δ.UGT8.4 and MCF.7 cells transfected with siRNA targeting P53 mRNA. The activities of the promoters cloned into the pGL3 basic vector were measured using the dual-luciferase reporter assay system. The bars represent the average luciferase activities compared to the control pGL3 vector. (**D**) The stability of Bcl-2 mRNA in MDA.Δ.UGT8.4 cells transfected with siRNA targeting P53 mRNA. Cells were treated with actinomycin D for 2–6 h, and after indicated time points, the Bcl-2 mRNA expression levels, which were normalized against β-actin mRNA, were evaluated through qPCR. MDA.Δ.UGT8.4 cells not treated with actinomycin D were used as calibrator samples. The results are shown as the percentage of the control (0 h). MDA.Δ.UGT8.4/siTP53 and MCF7.C/siTP53—BC cells transfected with siRNA directed against P53 mRNA; MDA.Δ.UGT8.4 and MCF7.C—control BC cells transfected with scrambled siRNA. Data represent the mean ± SD of three replicates from two independent experiments. Statistically significant differences (* *p* < 0.1; ** *p* < 0.01; *** *p* < 0.001). The original western blot of Figure 6A is in Appendix A.

**Figure 7 cancers-16-00389-f007:**
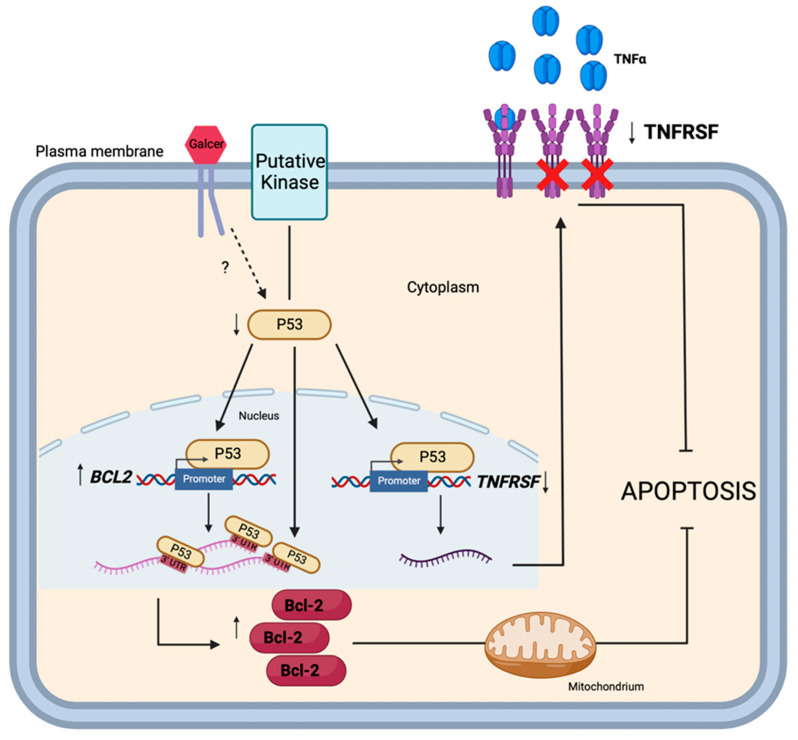
Diagram showing the putative regulatory role of GalCer in regulation of TNFRSF1B, TNFRSF9, and BCL2 gene expression that results in resistance of BC cells to apoptosis (Created with BioRender.com (accessed on 1 February 2023)). As hypothesized, GalCer modulates the activities of a signaling pathway/pathways which leads to decreased expression of P53. Depending on the cellular context, P53, acting as a transcription factor, upregulates expression of the BCL2 gene and downregulates the expression of TNFRSF1B and TNFRSF9 genes at the transcriptional level, or P53 acts as an mRNA-binding protein (RBP) and upregulates expression of Bcl-2 protein at the post-translational level. ↑—upregulation, ↓—downregulation, ?—unknown signaling pathway.

## Data Availability

The data that support the findings of this study are available from the corresponding author upon reasonable request.

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
