# Peer review of "Galactosylceramide Upregulates the Expression of the BCL2 Gene and Downregulates the Expression of TNFRSF1B and TNFRSF9 Genes, Acting as an Anti-Apoptotic Molecule in Breast Cancer Cells"

_cancers, 2024, doi:10.3390/cancers16020389_

Round 1

Reviewer 1 Report (Previous Reviewer 2)

Comments and Suggestions for Authors

Cancers--2768508_comments

The manuscript entitled “Galactosylceramide upregulates the expression of  BCL2 Gene and downregulates the expression of TNFRSF1B and TNFRSF9 genes, acting as an ani-apoptotic molecules in Breast Cancer Cells” by Jaroslaw Suchanski et al. was re-submitted to Cancers for possible consideration to publish. “anti-apoptotic” instead of “ani-….”. Keywords were not enough.

Both “simple summary” and “abstract” were too tedious. “Simple Summary” was not simple but too long. “Abstract” was too long and almost half of the introduction. Please shorten the length of both.

Line 326, “days” should be corrected into “day” or “Day”.

The font and size of texts in the figures were getting better.

Could it be possible that authors upload a revised manuscript without the revision labelings.

Major revision is suggested on the current uploaded version.

Comments on the Quality of English Language

Minor editing of English language required

Author Response

Reviewer 2 Report (New Reviewer)

Comments and Suggestions for Authors

The manuscript by Suchanski et al details the role of GalCer in breast cancer chemoresistance. This is a very well written and detailed manuscript that is timely and addresses a critical need in the field. 

A couple of minor issues should be addressed prior to publication: 

1. For the T47D model, the cells with altered UGT8 need to be shown to have changes in production of GalCer. 

2. On line 294, the authors compare the control cell lines to the parental cell lines, but these data are not shown. 

3. In section 3.2, the authors discuss the primary tumor results of the in vivo study. While they mention in the abstract/intro that GalCer is involved in primary tumor development, they do not see that here. It would be interesting to know why there is a discrepancy. 

4. The results in Figure 3B are not compelling for the TNFRSF1B in the MCF7 cell lines. Please confirm. 

Comments on the Quality of English Language

English quality is good. Only minor editing is needed. 

Round 2

Reviewer 1 Report (Previous Reviewer 2)

Comments and Suggestions for Authors

01.  I saw the raw images of the western blotting in Figs. 1, 2, 3 and 6. And the other two repeats were not shown for figures mentioned above. I wonder are all three WB repeats look the same or in similar tendency?

02. Figure 2C was too large but the rest parts were small. I recommend authors to reformat Fig. 2. Besides, under the layer of Figure 2A, there were again having (I) and (II), why not put as Panel A and Panel B in Fig. 2? Then revise Fig. 2B to 2C and then Fig. 2C as 2D.

03.  In discussion, from line 583 to line 670, the first two superlong paragraphs are so ling. Both are too long. It is impossible for the entire long paragraph completely focused on single point with single layer of the meaning. It must have a way to separate into 2 or three parts with some reasonable length each.

04. In the WB image, there was no arrow or line to indicate the position/MW, next to the membrane.

05.  The authors added a sentence of “The APC is financed by WrocÅ‚aw University of Environmental and Life 744 Sciences.” In the fundings section. Why? It was the same University and just a repeat of the same affiliation of the authors on Page One.

06.  In the graphical abstract, UGT8 and UGT8 were not shown. Compared to the terms used in the main texts, both P53 and p53 both used; BCL2, Bcl2 and Bcl-2 were used.

07. Reference 44#, was incomplete.

Comments on the Quality of English Language

some minor revision can be made on the English writing of the manuscript.

Round 3

Reviewer 1 Report (Previous Reviewer 2)

Comments and Suggestions for Authors

My questions have been answered and major issues have been addressed. 

I have no furthers questions. 

This manuscript is a resubmission of an earlier submission. The following is a list of the peer review reports and author responses from that submission.

Round 1

Reviewer 1 Report

Comments and Suggestions for Authors

The paper shows that UGT8, required for the synthesis of Galcer, modulates the response of breast cancer cells to chemotherapy by regulating the expression of apoptosis-related proteins. The paper is well written, however the relevance of UGT8 in resistance to chemio in BC has been already shown by the authors in a previous publication and the novelty is more on the mechanisms. A major limitation of the work is that the authors don’t show the effect of UGT8 modulation on other GSLs, even if they have the techniques required for this analysis. Modulation of UGT8 is indeed expected to change the expression of other GSLs not only of Galcer, also in breast cancer (BC). UTG8 expression for example activates the sulfatide biosynthetic pathway affecting the behavior of BC cells (PMID: 29728441). Downregulation of UGT8 could support the ganglioside synthesis. It is not possible from the experiments to conclude that Galcer is directly involved in the effects described in the paper. Moreover, the involvement of p53 is not sufficiently demonstrated.

Major

The authors should show the effect of UGT8 modulation on neutral and acidic GSLs by TLC and orcinol staining (or equivalent) and adapt the results and discussion accordantly.

The authors show that UGT8 expression modulates the expression of several genes on the level of transcription and mRNA stability, however the mechanism is not clarified. The involvement of p53 is not clearly demonstrated. Moreover, the authors didn’t discuss the mutation status of p53. MCF-7 is p53 WT, while MDA-MB-231 has a pathogenic p53 mutation with a very high allele frequency. How can mutated p53 (loss of function) be the mediator of UGT8 modulation in MDA-MB-231?

GSL composition can influence the adhesion behavior of BC and therefore signal transduction and gene expression. The authors show that UGCT modulation has not impact on morphology and proliferative potential. What about impact on adhesion?

The text in several figures is too small

Reviewer 2 Report

Comments and Suggestions for Authors

The manuscript entitled “Galactosylceramide as a Modulator of Drug Resistance and Regulator of TNFRSF1B, TNFRSF9, and BCL2 Gene Expression in Breast Cancer Cells” by Jaroslaw Suchanski et al. was submitted to Cancers for possible consideration as a publication. The major goal of the manuscript aimed to uncover the modulation of galactosylceramide in breast cancer cells but most likely too much emphasize the regulation of several key genes at the transcriptional levels and the detection of the promoter activities. However, the results in this study showed the anti-apoptosis effect of UGT8 in cancer cells. Meanwhile as the authors mentioned several times, galactosylceramide (GalCer) increased both tumorigenicity and metastasis of breast cancer (BC) and served in drug-resistance (DR) cancer cells as an anti-apoptotic molecule.

I have several major concerns on the current manuscript.

1)      The title was not clear or good and truly not fit with the main contents of the manuscript. By the way, this title showed no opinions. Did galactosylceramide (GalCer) show positive or negative effects on the drug resistance? Did GalCer up regulate or down regulate the mentioned genes? Not reflected in the title directly. The authors were not saying those in the title. Readers must read about the manuscript in order to know the answers to questions.

2)      I am curious about the co-presence of “Simple Summary” and “Abstract” on the Page 1. Are they parallelly required for and listed in each submitted manuscript?

3)      Three of four keywords were derived directly from the title of the current manuscript. In my mind, that was not generally accepted.

4)      It is quite confusing about the reason for the anti-apoptosis effect in cancer cells after overexpressing UGT8. They are caused by the accumulation of galactosylceramide (GalCer) or the alteration of protein levels of UGT8 in cancer cells? The author did not show any correlations between these apoptosis related genes and galactosylceramide (GalCer).

5)      And the underlying mechanism of galactosylceramide (GalCer)-dependent activation of p53 is also unknown.

6)      Regarding the western blotting results, for all the GAPDH in cancer cells, the bands for GAPDH were rather unstable or not crystal clear, with variations very much from one blot to the other blot, as other peer reviewers may notice this point.

Minor issues:

1)      “2.4. SDS-PAGE and western blotting” was incorrect therefore it should be revised.

2)      “2.5. Purification of neutral glycolipids and thin-layer chromatogram” should be separated.

3)      Did all the western blots performed only once in this study? If it were true, the results need careful repetition.

The quality of the data and main finding of this manuscript does not fully meet the basic criterion of publications on Cancers. Substantial revisions probably are necessary for re-consideration on the present manuscript. So, I recommend rejecting the manuscript unless the titles and objects are revised.

Comments on the Quality of English Language

English was good.

Reviewer 3 Report

Comments and Suggestions for Authors

J. Suchanski et al. reported that galactosyl ceramide (GalCer) increased the tumorigenicity and metastatic properties of breast cancer (BC) cells as well as their resistance to anti-cancer drugs acting as anti-apoptotic molecule. They demonstrated that GalCer specifically downregulated the level of pro-apoptotic TNFRSF1B and TNFRSF9 genes and increased the level of the anti-apoptotic BCL2 gene expression.

However, this transactivation mode was not well characterized to identify the controlled elements, although the author suggested the possible cis-elements like CREB1, NFkB and p53 binding sites in each TNFRSF1B, TNFRSF9 and BCL2 promoters in response to GalCer response, but they did not characterized in detail. Thus, the author needs to characterize the key elements in two positive controls of TNFRSF1B, TNFRSF9 promoters and negative control of BCL promoter by a series of deletion and mutation analysis of each promoter.

In addition, the author suggested the control of Bcl2-transcript stability, possibly through p53. However, they did not show any direct evidence of these interpretations as p53 molecule.

Moreover, the authors proposed that the inhibition of UGT8 by knockout UGT gene or using chemical inhibitors of UGT8 enzymes would increase cancer cell sensitivity to conventional chemotherapy. However, there is no mechanistic analysis of UGT induced modification. This, these works are still preliminary for publication.

In addition, MDA-MB-231 cells and T47D cells have the mutated p53 (Hum Mutat 19: 607–614, 2002; Br. J. Cancer, 101, 1606-1612, 2009), but MCF7 has a wild type p53 (Clin Cancer Res 7: 2114–2123, 2001) and 4T1 is p53 null (Cancer Gene Therapy, 12, 427–437, 2005). Thus, it is required to characterize the WT p53 and mutant p53 in each cell in the transcriptional and post-transcriptional regulation of these target gene promoters in response to GalCer. This p53-status is critical for evaluation of the results of each experiment.

Without these data it is hard to accept the present form of this manuscript.

[Major points]

1.      In this article, authors described the GalCer expressed tumors with more metastatic properties and tumorigenicity. However, the authors did not show the properties of these tumor under the treatment of DOX or other chemotherapeutic agents with or without the expression of GalCer, or UGT8 gene. It is required for the authors to confirm the properties of these tumors and characterize the role of GalCer effect on the tumors.

2.      In this article, authors focused only on the apoptosis effect driven by GalCer depletion. It is recommended to investigate other types of programmed cell death (ex. Pyroptosis, programed necrosis, ferroptosis,  autophagy, etc..) in order to establish the role of GalCer in cancer progressions.

3.      In this article, the authors showed the role of GalCer to modulate BCL-2 gene by correlation of p53 expression and knock out of UGT8 gene was positively correlated. But the role of GalCer in cancer is yet to be confirmed, it is difficult to conclude GalCer as a modulator of drug resistance. It is recommended to include more evidence such as the interaction data of GalCer, western blot or qPCR of activation pathway, analysis of database of GalCer expression with TNFRSF1B, TNFRSF9 in patients.

Minor points:

1.      In this article, authors selected three commonly used chemotherapeutic agents: DOX, PAX, and CDDP. However, these three agents are very different in their mechanisms of chemotherapeutic functions. But all showed significance in effect of GalCer depleting experiments. Can the author demonstrate the role of GalCer in the pathway, or in the mechanism of these chemotherapeutic agents?

2.      In Figure 6B: Please show all explanations of each schematic modeling. The explanation of some scheme is missing.

Comments on the Quality of English Language

Please ask the specific company for English editing.